# Anti-Cancer Drugs: Trends and Insights from PubMed Records

**DOI:** 10.3390/pharmaceutics17050610

**Published:** 2025-05-04

**Authors:** Ferdinando Spagnolo, Silvia Brugiapaglia, Martina Perin, Simona Intonti, Claudia Curcio

**Affiliations:** 1Department of Molecular Biotechnology and Health Sciences, University of Turin, Piazza Nizza 44bis, 10126 Turin, Italy; ferdinando.spagno57@edu.unito.it (F.S.); silvia.brugiapaglia@unito.it (S.B.); martina.perin@unito.it (M.P.); simona.intonti@unito.it (S.I.); 2School of Advanced Defence Studies, Defence Research & Analysis Institute, Piazza della Rovere 83, 00165 Rome, Italy; 3Defense Institute for Biomedical Sciences, Via Santo Stefano Rotondo 4, 00184 Rome, Italy

**Keywords:** PubMed, meta-analysis, anti-cancer drugs, combined therapy

## Abstract

**Background:** In recent years, there has been an exponential growth in global anti-cancer drug research, prompting the necessity for comprehensive analyses of publication output and thematic shifts. **Methods**: This study utilized a comprehensive set of PubMed records from 1962 to 2024 and examined growth patterns, content classification, and co-occurrence of key pharmacological and molecular terms. **Results**: Our results highlight an exponential rise in publications, with an annual compound growth rate of over 14%, influenced by advancements in digital knowledge sharing and novel therapeutic breakthroughs. A pronounced surge occurred during the COVID-19 pandemic, suggesting a sustained shift in research dynamics. The content analyses revealed a strong emphasis on classical chemotherapeutic agents—often studied in combination with targeted therapies or immunotherapies—and a growing focus on immune checkpoint inhibitors and vaccine platforms. Furthermore, co-occurrence networks indicated robust links between chemotherapy and supportive care, as well as emerging synergies between immuno-oncology, precision medicine approaches. **Conclusions**: Our study suggests that while novel modalities are reshaping treatment paradigms, chemotherapy remains central, underscoring the value of integrative regimens. This trend toward personalized, combination-based strategies indicates a transformative era in oncology research, where multidimensional data assessment is instrumental in guiding future therapeutic innovations.

## 1. Introduction

Cancer is a leading cause of death worldwide, driving ongoing research into anti-cancer drugs and treatment advancements [1]. Over recent decades, therapeutic strategies have evolved from traditional cytotoxic agents like platinum-based compounds and taxanes [2] to targeted therapies that inhibit molecular drivers (e.g., EGFR, HER2, BRAF) [3,4] and cancer vaccines [5]. Immunotherapy, especially immune checkpoint inhibitors (ICIs) targeting CTLA-4 and PD-1/PD-L1, has revolutionized treatment by using the immune system to achieve durable remissions.

Despite these advancements, chemotherapy remains essential, especially in resource-limited settings and as part of combination regimens. Drug resistance, driven by tumor cell mechanisms like ATP-binding cassette transporters and altered apoptotic pathways, remains a significant challenge. As a result, research focuses on overcoming chemoresistance, improving drug delivery, and personalizing treatments with biomarkers.

Despite these advances, classical chemotherapy remains essential, especially in low-resource settings and as a base in combination therapies [6].

Research is intensifying on overcoming chemoresistance, improving targeted delivery, and using biomarkers to personalize treatment [6]. Preventive vaccines for human papillomavirus (HPV) and hepatitis B virus (HBV) have shown success in reducing infection-related cancers, while therapeutic vaccines—peptide-based, dendritic cell, and mRNA—are being developed to stimulate tumor-specific T-cell responses [7,8,9,10,11,12].

Combining ICIs with oncolytic viruses or cancer vaccines offers promise for overcoming resistance [13,14]. Advanced delivery systems like liposomes and nanoparticles enhance chemotherapy’s effectiveness and reduce toxicity [7]. These strategies reflect a shift toward precision oncology, integrating immunotherapy, genomics, and personalized care to improve outcomes.

This study uses PubMed data from 1962 to 2024 to analyze trends in cancer drug development through bibliometric and semantic methods. It highlights the rise in targeted and immune-based therapies alongside chemotherapy, revealing key challenges like resistance and side effects, and providing a framework to guide clinical decision-making.

## 2. Materials and Methods

We retrieved PubMed records from 1962 to early 2025 using the ENTREZ API, targeting articles that included keywords and substances related to anti-cancer drugs.

To quantify the number of articles published, the Compound Annual Growth Rate (CAGR) was calculated, which measures the constant growth rate over time using the following formula:(1)$$CAGR={\left(\frac{V_{f}}{V_{i}}\right)}^{\frac{1}{n}}-1$$
where Vf = 3497 (publications in 2024); Vi = 1 (publications in 1962); and n = 62 years (from 1962 to 2024). This method was selected for its ability to estimate average annual growth by smoothing out year-to-year fluctuations, thereby offering a reliable representation of overarching trends, even in the presence of occasional inconsistencies within the dataset.

After collecting the full XML metadata for each PMID, we processed the data in R (version 4.4) to perform bibliometric, textual, and classification analyses. Our scripts parsed all records, transforming them into structured tables for further exploration. Publication counts were then aggregated by year, country of origin, and number of authors, enabling trend analyses and descriptive statistics. For content assessment, we grouped relevant terms into categories (e.g., chemotherapy agents, immune checkpoints, targeted therapies) to visualize their temporal evolution. We also extracted trigrams from the *NameOfSubstance* field to capture common three-term co-occurrences, normalized the data based on each trigram’s maximum recurrence, and examined shifts in publication focus before and during the COVID-19 pandemic. All intermediate tables, scripts, and summaries are available upon request or in the Appendix A.

The classification of terms was determined through a consensus-based methodology. Initially, all trigrams derived from the *NameOfSubstance* field were identified and ranked. Subsequently, the top 250 distinct trigrams were selected and each individual *NameOfSubstance* was assigned to the most appropriate class based on the authors’ consensus.

Eleven anti-cancer classes were identified:**Apoptosis and Cell Death**: Proteins and mechanisms involved in programmed cell death (e.g., caspases, Bcl-2, p53), crucial in tumorigenesis and cancer treatment.**Chemotherapy Agents**: Cytotoxic drugs that damage DNA, inhibit nucleotide synthesis, or interfere with cell division (e.g., alkylating agents, antimetabolites, platinum compounds).**Growth Factor Receptors**: Characterized primarily by their direct modulation of oncogenic signaling pathways controlling angiogenesis (e.g., VEGF/VEGFR-2/KDR), cell proliferation, differentiation, and survival (e.g., EGFR, ERBB2/HER2, IGF-I, FGF, TGF-β).**Immune Checkpoint**: Defined specifically by their immunomodulatory action, reversing tumor-induced immune suppression through the targeted inhibition of checkpoint molecules (e.g., PD-1/PD-L1 axis, CTLA-4), thus restoring antitumor T-cell activity and significantly improving patient outcomes in various tumor types.**Cytokines**: Immune-modulating cytokines (e.g., interferons, interleukins) used as adjuvants to enhance immune response against cancer.**Anti-inflammatory Drugs**: NSAIDs and COX inhibitors (e.g., aspirin, celecoxib) with potential roles in cancer prevention and treatment through inflammation modulation.**Monoclonal Antibodies**: Distinguished by their high antigen specificity, representing a therapeutic platform engineered to selectively bind and neutralize defined tumor-associated antigens across diverse malignancies (e.g., CD20, CD19) and advanced antibody formats (e.g., immunoconjugates, immunotoxins, bispecific antibodies).**Phytogenic Agents**: Plant-derived compounds (e.g., alkaloids, flavonoids) with anti-cancer potential.**Supportive Therapies**: Non-anti-cancer drugs (e.g., analgesics, antiemetics) that manage cancer-related symptoms and improve patient quality of life.**Targeted Therapies**: Molecules targeting key signaling pathways (e.g., tyrosine kinase inhibitors, mTOR inhibitors), offering more precise treatments than chemotherapy.**Tumor Suppressors**: Proteins (e.g., p53, PTEN) that regulate cell growth, often inactivated in cancer, with research focused on reactivating or targeting these pathways.Detailed information on the topics in each class is available in Section A.1, and an overview of the drugs is shown in Figure 1.

Although “monoclonal antibodies” represent a broad and structurally diverse therapeutic category—including agents targeting both growth factor receptors and immune checkpoint molecules—we identified distinct research trends that justify treating ICIs as a separate class. Given the clinical significance and mechanistic specificity of therapies targeting growth factor pathways and immune checkpoints, we found it scientifically appropriate to categorize these independently. As a result, we limited the “monoclonal antibodies” category to those targeting well-defined tumor-associated antigens with established clinical applications. This distinction reflects both structural and functional differences. Specifically, agents in the “Growth Factor Receptor” category target key signaling pathways involved in angiogenesis and tumor cell proliferation, while “Immune Checkpoint” therapies are defined by their capacity to restore suppressed immune responses, leading to improved outcomes across a wide range of cancers. Figure 2 illustrates a Venn diagram depicting the potential overlaps among the three classes of terms, each of which includes compounds associated with more than 50 PMIDs, with detailed list provided in the Appendix A).

## 3. Results

### 3.1. General Trend in Scientific Production (1962–2024)

The analyzed dataset shows a right-skewed distribution with significant variability in scientific publications from 1962 to 2024. The mean number of publications is 733.51, but the median is much lower at 245, indicating that most years had fewer publications, with occasional high peaks (Figure 3, Table 1). The data dispersion (1037.40 ± 3496) and interquartile range (933.5) reflect a large difference between the first and third quartiles. A skewness index of 1.52 confirms a long right tail, and a kurtosis near 1 suggests a slightly flatter distribution compared to a normal one. This variability is likely driven by factors such as technological advancements, regulatory changes, or specific events that caused spikes in research activity.

The number of scientific publications has grown exponentially, with a notable rise in recent decades. From just one article in 1962, publications reached 3497 in 2024, with a Compound Annual Growth Rate (CAGR) of 14.05%. This growth is driven by factors such as increased global research investment, expanded digital access, greater international collaboration, open-access journals, and advancements in scientific technology.

### 3.2. Geographical Scientific Contribution in Publication

The distribution of scientific publications shows a global disparity, with a few countries dominating research output (Figure 4). Developed nations, particularly the United Kingdom (14,605 publications), United States (10,677), and Switzerland (5581), lead in scientific publications. Other countries like the Netherlands (4501) and Germany (1895) also contribute significantly. In contrast, many countries publish fewer than 50 articles (Figure 4).

The number of authors per article reflects research complexity and collaboration levels. From 1969 to 2024, the median number of authors typically ranged from 2 to 7, with most articles involving fewer than 10 authors (Section A.2). The interquartile range shows smaller teams dominate publishing. However, after 2015, there was an increase in outliers (articles with over 50 authors), coinciding with large-scale global initiatives, particularly following the OMICs revolution [15,16,17].

### 3.3. Anti-Cancer Drug Classification and Grouped Analysis

To enhance the analysis of anti-cancer drug research, we focused on the *NameOfSubstance* field in PubMed records. By analyzing these fields, we identified key relationships and trends in the research area. The analysis was conducted at three levels, single-element, two-element, and three-element groups, enabling us to achieve the following objectives:*Quantitative tracking:* Measured the frequency of cited elements, revealing their temporal evolution and role in shaping scientific discourse.*Link analysis:* Examined relationships between elements, uncovering dynamics within and across research themes.*Article classification:* Categorized articles into general topics and niche areas, providing insight into the research landscape.

This approach revealed both macro- and micro-level trends, helping us understand the thematic structure of anti-cancer drug research and the evolution of scientific ideas.

The *NameOfSubstance* field was chosen for its specificity and consistency in identifying pharmacological compounds (Table 2), making it ideal for bibliometric analysis. Unlike other indexing fields, it avoids redundancy and ensures reproducibility through standardized terms. This enables effective tracking of research trends, emerging drug combinations, and shifts in pharmaceutical focus—particularly in response to major events like **COVID-19**. It also supports hierarchical classification of therapies, enhancing insight into the evolution of treatment categories. Over the past three decades, oncology publications have increased significantly, with recent plateaus suggesting a transition toward more complex, personalized, and combination-based therapies, as shown in Figure 5 and detailed in Section A.3.

After classification, a first global assessment was conducted to analyze the relationships within categories by means of an undirected weighted graph, with *Nodes* representing drug-related categories, *Edges* representing the relationships between categories established when two categories co-occur in the same PubMed article, and *Weights* indicating the number of shared PubMed IDs (PMIDs) between two categories, meaning the strength of the association (Figure 6).

Figure 7 illustrates the shifting landscape of cancer treatment, showing a transition from traditional chemotherapy to more targeted and immune-based approaches. Chemotherapy Agents exhibit the strongest relationships with Phytogenic Agents, Supportive Therapies, and Targeted Therapies, reflecting their central role in both established and evolving regimens. Moderate connections between Monoclonal Antibodies, Immune Checkpoints, and Targeted Therapies indicate growing interest in immunotherapy. Weaker links, such as those involving Tumor Suppressors, suggest more specialized research focus. Relationship strengths between categories are quantified in the **category_pairs.tsv** file. Emerging patterns highlight the increasing integration of biomarkers and growth factor receptors, and the expanding but still developing role of immunotherapies. Across all categories, **Supportive Care** remains vital for managing side effects and enhancing treatment tolerability. Figure 7 provides an overview of these trends based on *NameOfSubstance* trigram frequencies.

### 3.4. COVID-19 Pandemic Impact on Research Directions

To assess the impact of the COVID-19 pandemic on scientific publications, we compared the observed data to a baseline representing expected publications in a normal year (Section A.4). Key trends include the following:2020 (Surge +24.64%): Publications increased sharply to 3197 articles, with a deviation of +335.9%, indicating a major shift in research activity.2021 (Peak +8.20%): Publications rose to 3459 articles, reaching a maximum deviation of +534.8%, marking the peak of pandemic-driven research.2022–2023 (Plateau 0.37–0.72%): Growth slowed but remained high, with a deviation above +375%, indicating a new steady-state.2024 (Stabilization −0.97%): Publications slightly declined to 3463, but still remained above historical trends, suggesting a structural shift.

The COVID-19 pandemic significantly impacted the quantity of scientific publications. To assess its effect on anti-cancer drug research, we analyzed trigrams trends. We calculated the annual average PMID counts for relevant trigrams, normalized the data based on each trigram’s maximum occurrence, and plotted these trends to visualize shifts in research focus over time (Figure 8). The normalized values represent the average proportional mention intensity of trigrams across PubMed articles from 2010 onward.

From 2010 to 2019, there was a steady increase, peaking at 0.324 in 2015. During the pandemic (2020–2024), there was a brief rebound to 0.292 in 2020, likely due to oncology studies incorporating pandemic-related topics like immunosuppression, drug repurposing, and antiviral synergy. However, this was followed by a decline (0.155 in 2022 and approximately 0.06 by 2023–2024). Possible reasons include the reallocation of research due to pandemic restrictions, a shift toward mRNA therapeutics and vaccines, and funding changes that deprioritized certain areas [18,19,20]. Additionally, *NameOfSubstance* trigrams were used to assess shifts in specific research areas (Figure 9).

The datasets show a surge in references to mRNA- and RNA-based platforms, such as “mRNA vaccines” and “RNA-based therapies”, driven by the success of COVID-19 vaccines [21]. Growth in trigrams like “ICIs” and “CAR T-cells” reflects rising interest in advanced immuno-oncology [22,23], while increased mentions of supportive and palliative care relate to heightened awareness of complications in immunocompromised patients [24]. Short-term spikes were observed for repurposed COVID-19 drugs, including tocilizumab and baricitinib. Conversely, mentions of older chemotherapy terms like “alkylating agents” and “antimetabolites” declined, reflecting a shift toward immunotherapies and targeted treatments [2]. The pandemic also slowed niche research due to lab shutdowns and redirected resources [19], and Phase III trials faced enrollment challenges, lowering citation rates for complex regimens [6].

Looking ahead, big data analytics and real-world evidence are expected to reshape treatment strategies and influence publication trends [16,24]. As detailed in Section A.5, three key areas stand out in anti-cancer drug research: Chemotherapy Agents, for their foundational role; Immune Checkpoints, for their transformative therapeutic impact; and Vaccines, particularly those accelerated by the COVID-19 response [21,22,23]. While newer modalities gain traction, cytotoxic chemotherapy remains integral, with ongoing refinements in dosing, scheduling, and combinations with biologics (Section A.5
Figure A2).

### 3.5. Publication Trends

#### 3.5.1. Immune Checkpoint

The Immune Checkpoint category is central in cancer therapy, connecting with various other categories such as monoclonal antibodies, targeted therapies, and traditional chemotherapy. Its effectiveness is influenced by factors like inflammation, drug resistance, and tumor suppressor gene status (Figure 10). Understanding these intricate relationships allows for more strategic combination therapies and better-informed clinical decisions.

An overview of how the Immune Checkpoint category functionally and therapeutically interacts with other major categories is shown in Table 3, based on the classification data (file classification_NameOfSubstance.txt for NameOfSubstance classification) and the output from graph generation Appendix A, alongside established scientific evidence in oncology and immunology.

A total of 243 articles referencing ICIs in the *NameOfSubstance* field reveal evolving trends in immunotherapy, including safety, biomarker-driven personalization, and combination strategies. Early focus on PD-1/PD-L1 and CTLA-4 has expanded to include LAG-3, TIGIT, and PVRIG, reflecting broadening checkpoint targets. From 2019 to 2025, ICI applications have grown across both immunogenic cancers (e.g., melanoma) and traditionally resistant types like KRAS-mutant pancreatic cancer and microsatellite-stable colorectal cancer, with synergistic effects observed in non-small cell lung cancer (NSCLC) treated with EGFR inhibitors and ICIs [25,26].

Recent studies emphasize combinatorial approaches, pairing ICIs with tyrosine kinase inhibitors, anti-angiogenics, and epigenetic modulators (e.g., HDAC and DNA methylation inhibitors) to overcome resistance and improve efficacy [27,28,29,30,31,32]. Persistent efforts focus on resistant tumors such as MSS colorectal cancer and KRAS-driven malignancies [33,34]. Innovative drug delivery systems, like nanocomposite gels and liposomal carriers, aim to localize immune effects and reduce systemic toxicity [35,36].

Classic drugs—cisplatin, paclitaxel, sorafenib, trastuzumab, and others—remain central, often featured in combination trials, underscoring the role of immunomodulatory synergy between traditional and immune-based treatments. The presence of analgesics, antacids, and metabolic modulators suggests an emerging interest in how supportive medications influence the tumor immune microenvironment. Lower-frequency mentions in Q3/Q4 may represent early-stage agents or niche approaches with future clinical potentials.

#### 3.5.2. Cancer Vaccine Therapies

Cancer vaccines are rapidly emerging as a pivotal area in immuno-oncology, aimed at providing durable antitumor immunity. These vaccines, which can complement existing therapies like ICIs, targeted therapies, and chemotherapy offer significant promise for advancing cancer treatment strategies. A filtered dataset of 306 cancer vaccine-related records (detailed in Appendix A) provides valuable insights into this evolving field.

The dataset distinguishes between Prophylactic Vaccines, designed to prevent infection-related cancers (e.g., HPV vaccines for cervical, anal, and oropharyngeal cancers [10] and HBV vaccines to reduce liver cancer risk [9]), and therapeutic vaccines, aimed at treating existing cancers by enhancing the immune response to tumor antigens [4,11,12]. ICIs, like anti-PD-1 and anti-CTLA-4, complement therapeutic vaccines by expanding tumor-specific T-cell populations and preventing immune exhaustion [23]. Chemotherapy can enhance vaccine efficacy through immunogenic cell death, which releases tumor antigens and boosts immune responses, while radiotherapy further modulates the tumor microenvironment [37]. Targeted therapies, such as EGFR or VEGF blockade, also aid in vaccine effectiveness by increasing antigen release and immune infiltration.

Emerging platforms like mRNA-based vaccines and dendritic cell vaccines (as noted in vaccine_data.tsv) are gaining traction, with early-phase clinical trials showing promising immune responses. Oncolytic virus-based vaccines, like T-VEC, function as virotherapy while enhancing antitumor immunity through tumor antigen expression [13].

The dataset also highlights three principal clinical challenges: tumor heterogeneity, which requires personalized or multi-antigen approaches [14]; immunosuppression, where combining vaccines with ICIs or immunomodulatory agents is crucial; and limited immunogenicity of self-derived tumor antigens, necessitating careful adjuvant and epitope selection [38]. The data reflect an expanding pipeline of neoantigen vaccines, dendritic cell-based vaccines, and next-generation RNA vaccines, with therapeutic vaccines continuing extensive clinical testing.

## 4. Discussion

Our comprehensive bibliometric and content-based assessment of PubMed records highlight the swift evolution of anti-cancer drug research over multiple decades.

While we have meticulously employed rigorous bibliometric methodologies to mitigate potential biases, it is important to acknowledge inherent limitations. Specifically, certain influential factors and phenomena remain elusive in purely bibliometric analyses of PubMed records. Although institutional affiliations such as universities, institutes, or foundations, as well as broad categories of funding sources, are explicitly identified during the manuscript submission process, numerous underlying drivers of research trends may remain hidden. Factors including parallel financing streams, opportunistic research initiatives by individual researchers or departments, and other transient or emergent influences could significantly impact research trajectories. Capturing these dynamics would necessitate alternative research methodologies and a distinct analytical focus oriented towards identifying and characterizing such “driving forces,” rather than solely quantifying trends from bibliometric datasets.

The analysis revealed a pronounced acceleration in research output during key inflection points, such as the onset of the COVD-19 pandemic. The exponential growth of publications reflects both technological and clinical advances, including the advent of high-throughput genomic platforms and the surge in immuno-oncology. In particular, our classification scheme, which organizes substances into categories like chemotherapy, immune checkpoint, targeted therapies, and phytogenic agents, highlights the expanding complexity of contemporary oncology. The results also confirm the ongoing importance of older cytotoxic regimens in resource-limited settings and as backbones for combination protocols.

Several findings stand out. First, classical chemotherapy agents, despite the rise in targeted and immune-based strategies, remain fundamental to many clinical protocols, illustrating the durability of well-established cytotoxic mechanisms. Second, immunotherapies have reshaped treatment paradigms with notable success in various tumor types, yet their broad adoption also raises issues of drug resistance, long-term toxicities, and heterogeneous response rates. Third, the persistent integration of biomarkers—circulating RNA, protein markers, and genomic variants—signals a collective effort to refine patient stratification and optimize therapeutic outcomes. Finally, supportive therapies addressing side effects and comorbidities emerge as crucial in extending survival and enhancing patient quality of life, especially in multi-drug combination regimens.

In the context of prior research, these trends mirror the widely acknowledged shift towards personalized medicine: we observe greater reliance on combination strategies that merge targeted agents, immunomodulators, and classical drugs, with robust supportive care to manage adverse events. This integrative paradigm aims to exploit multiple vulnerabilities within tumors while improving tolerability. The data further suggest that emerging areas, such as mRNA-based vaccines and advanced nanocarrier delivery systems, could drive the next wave of innovation, accelerating synergy with immunotherapies and enabling more precise drug deployment.

Looking forward, future research directions may revolve around three core themes: (1) deepening multi-omics profiling to better predict therapy responses and tailor regimens to individual tumor biology; (2) developing improved mechanisms to overcome or preempt drug resistance—particularly by targeting ABC transporters, modulating tumor microenvironments, or leveraging synthetic lethality approaches; and (3) adopting advanced data-driven methodologies, including real-world evidence and machine learning, to more rapidly optimize treatment sequencing and dosage. Taken together, these converging lines of inquiry are poised to redefine anti-cancer drug development, promoting a multifaceted and patient-centered approach that aligns discovery with evolving clinical needs.

Bibliometric data show a pronounced shift toward personalized cancer medicine, evidenced by the frequent co-occurrence of ICIs, and advanced drug delivery platforms such as nanocarriers and immunoconjugates. At the same time, high publication counts on supportive care therapies highlight an ongoing commitment to optimizing patient outcomes through toxicity mitigation and quality-of-life measures. Our classification scheme, based on the *NameOfSubstance* field, underscores the complex interplay among drug resistance mechanisms, growth factor signaling pathways, immunomodulation, and emerging vaccine technologies. Taken together, these findings indicate that future research will likely focus on synergy-oriented regimens—integrating established cytotoxins, new molecular targets, immune-based strategies, and innovative formulations.

The rise in biotechnology has led to increased investment in research and clinical trials involving monoclonal antibodies. In this context, a range of combinatorial strategies has been employed to strengthen the clinical evidence supporting ICIs and to promote their potential use in refractory cancers such as pancreatic cancer, as demonstrated both in this study and in previously published research [39,40,41].

## 5. Conclusions

By systematically mapping past and present trends, this study provides a foundation for anticipating how forthcoming scientific advances, global health events, and collaborative consortia will continue to reshape the anti-cancer drug landscape. Capturing these recurring trigrams across such diverse categories, we see a richly interlinked oncology field, where formerly siloed approaches—like chemotherapy, immunotherapy, and supportive therapies—converge into strategic, patient-centric regimens. This integration reflects a paradigm shift in cancer treatment, emphasizing a holistic approach where different therapeutic modalities are no longer viewed in isolation but rather as complementary components of a broader, more effective strategy. Future research will likely build on this convergence, leveraging multi-modal interventions that optimize treatment efficacy while minimizing adverse effects, ultimately driving improvements in patient survival and quality of life.

## Figures and Tables

**Figure 1 pharmaceutics-17-00610-f001:**
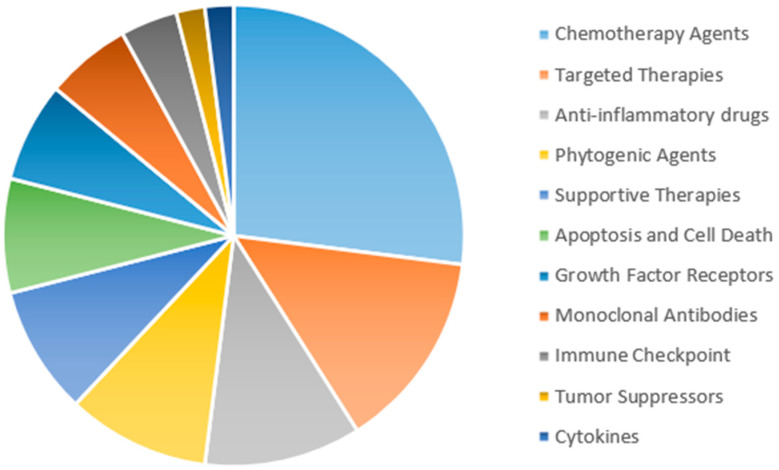
A pie chart representation of the *NameOfSubstance* distribution by category, illustrating the classification of substances related to anti-cancer drug research. Each colored section corresponds to a distinct category and the size of each section is proportional to the distribution of the corresponding terms.

**Figure 2 pharmaceutics-17-00610-f002:**
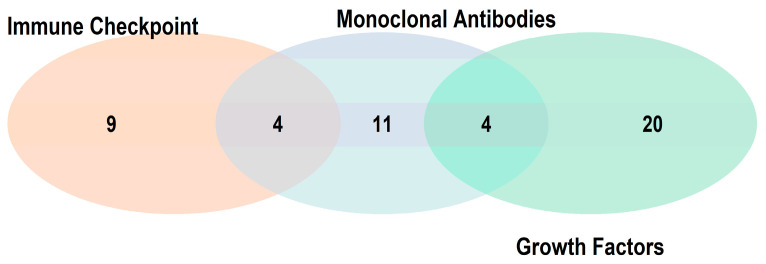
A Venn diagram depicting potential overlaps among the three identified classes (“Growth Factors,” “Immune Checkpoint,” and “Monoclonal Antibodies”), considering only terms counting more than 50 PMIDs. The intersection between *Growth Factors* and *Monoclonal Antibodies* includes trastuzumab, cetuximab, panitumumab, and bevacizumab. The intersection between *Immune Checkpoint* and *Monoclonal Antibodies* includes nivolumab, pembrolizumab, ipilimumab, and atezolizumab. No biologically significant overlap was identified between *Growth Factors* and *Immune Checkpoint*, nor among all three categories simultaneously.

**Figure 3 pharmaceutics-17-00610-f003:**
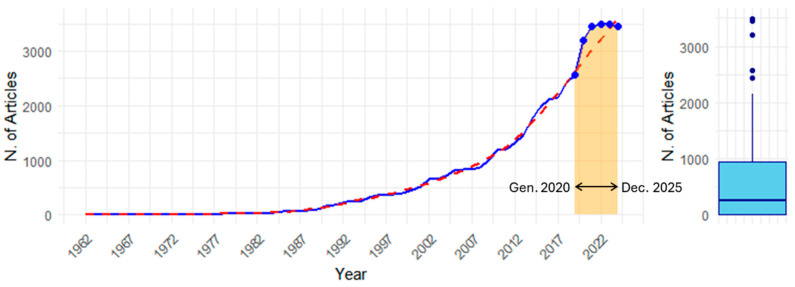
A time series of article counts and their distribution. The left line plot depicts the growth in the number of publications over the years, while the right boxplot shows the interquartile range, with the central line indicating the median, whiskers representing the data range, and points above representing outliers.

**Figure 4 pharmaceutics-17-00610-f004:**
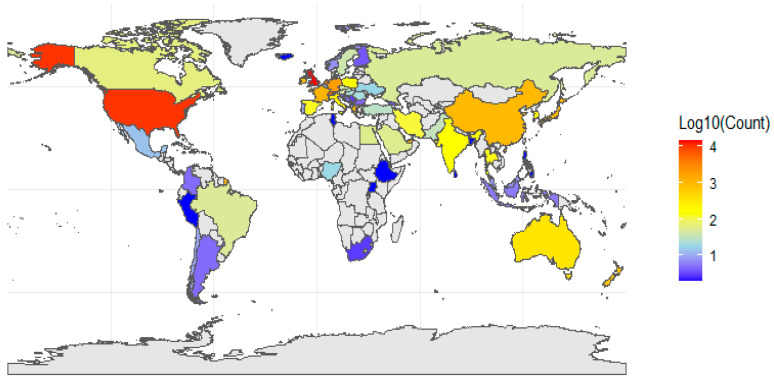
Geographical distribution of total published articles by editor countries, displayed on logarithmic scale to enhance visualization.

**Figure 5 pharmaceutics-17-00610-f005:**
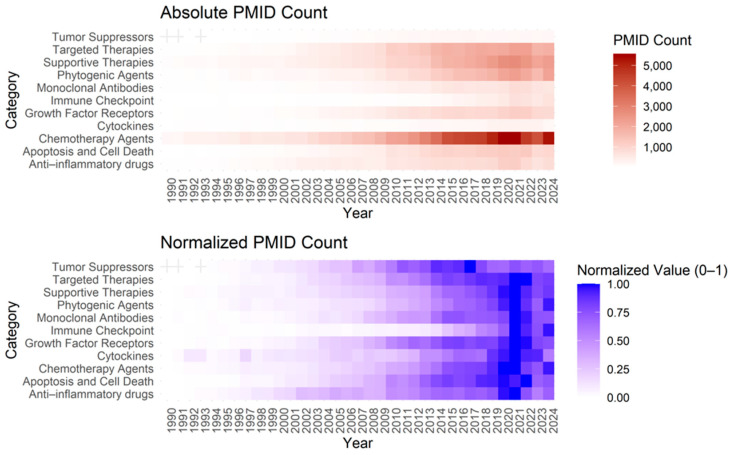
Heatmaps illustrating the temporal evolution of PMID counts across different categories of anti-cancer drug research. Top Panel (Absolute PMID Count): Raw number of PubMed-indexed publications per category from 1990 to 2024. Darker shades indicate a higher volume of research activity. Bottom Panel (Normalized PMID Count): Relative trend within each category, normalized between 0 and 1 for intra-category comparison purposes.

**Figure 6 pharmaceutics-17-00610-f006:**
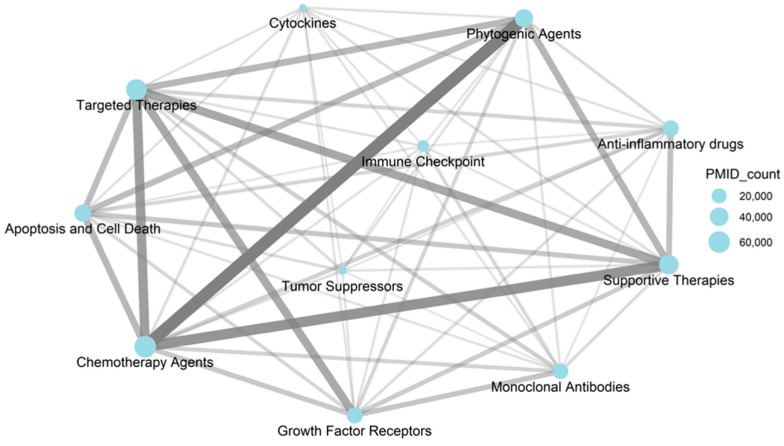
The relationships within the categories. Graph description: Nodes (Categories): Each node represents a category of drug-related research. The size of the node is proportional to the number of PubMed articles (PMID count) associated with that category. Larger nodes indicate categories with a higher number of related articles. Smaller nodes represent categories with fewer associated publications. Edges (Connections): The edges represent the co-occurrence of categories in the same PubMed articles. Thicker and darker edges indicate a stronger relationship (higher number of shared articles). Thinner and lighter edges signify weaker relationships with fewer shared publications.

**Figure 7 pharmaceutics-17-00610-f007:**
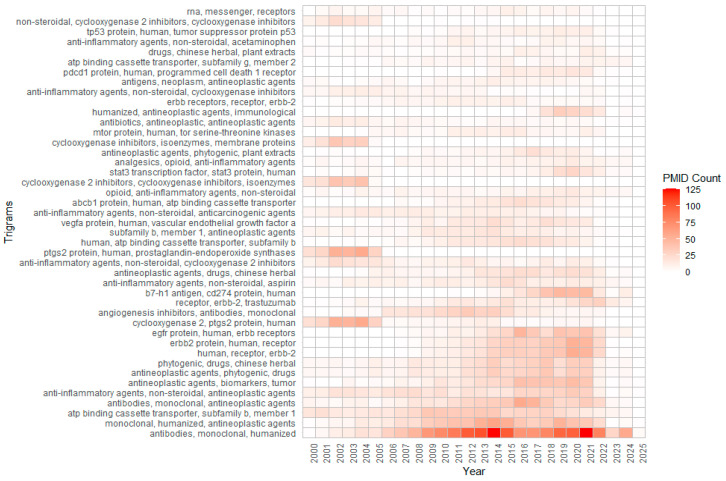
Recurring trigrams of anti-cancer drugs overtime. An interactive graph of all trigrams is available in the Appendix A.

**Figure 8 pharmaceutics-17-00610-f008:**
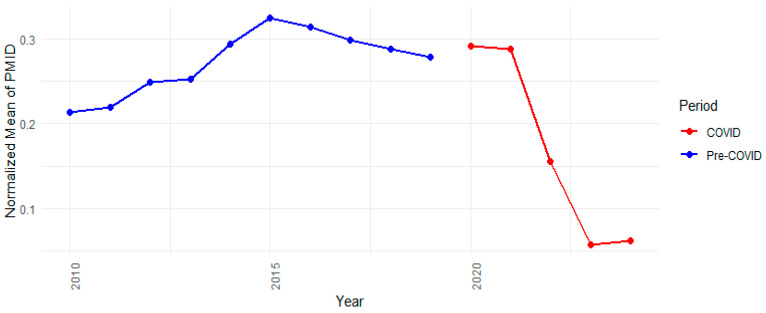
Trigrams and deviations trend during COVID-19 pandemic.

**Figure 9 pharmaceutics-17-00610-f009:**
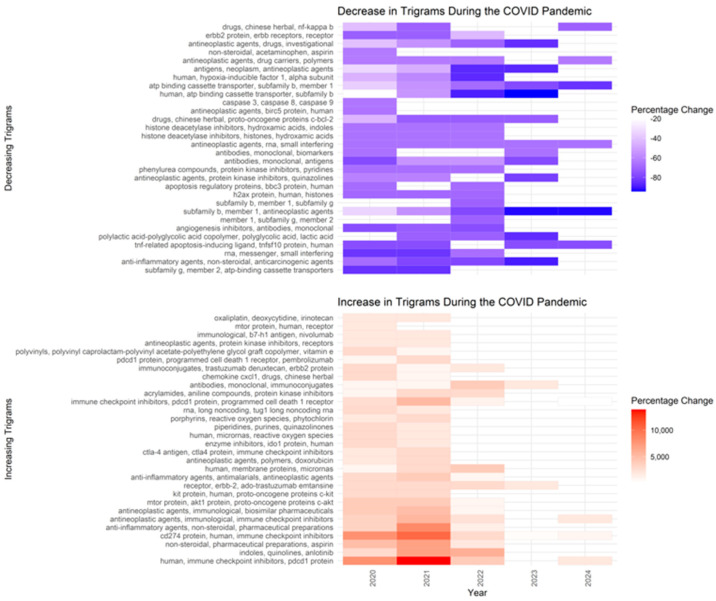
Heatmaps illustrating the percentage change in *NameOfSubstance* trigram counts during the COVID-19 pandemic wave. The top panel shows the 30 most decreased trigrams, while the bottom panel highlights the 30 most increased.

**Figure 10 pharmaceutics-17-00610-f010:**
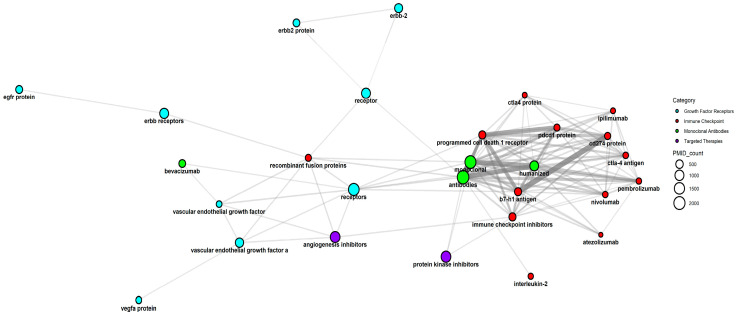
Relationships among Immune Checkpoint-related substances. Nodes represent different substances, with size proportional to their PMID count. Edges indicate co-occurrence within the same study. Stronger connections (darker edges) highlight frequent co-citations, suggesting stronger associations in the research literature. Categories are color coded as shown in the right-side legend.

**Table 1 pharmaceutics-17-00610-t001:** Descriptive statistics of number of scientific publications over time related to “*anti-cancer drugs*” PubMed prompt.

**min**	**MAX**	**mean**	**median**	**SD**	**variance**
1	3497	733.5079	245	1037.403	1,076,204
**Q1**	**Q3**	**IQR**	**range**	**skewness**	**kurtosis**
15	948.5	933.5	3496	1.515994	1.13817

**Table 2 pharmaceutics-17-00610-t002:** Overall statistics of PubMed records related to “*anti-cancer drugs*” PubMed prompt, analyzing distribution of PMID fields *NameOfSubstance*, *DescriptorName*, and *Keywords*.

	NameOf Substance	Descriptor Name	Keywords
TotalPMIDs	37,521	39,237	23,330
Total Unique Elements	14,456	13,105	47,107
Median Elements Per PMID	5	14	5
Max Elements	39	54	216
PMIDs With 1 Element	3280	5	17
PMIDs With 2 Elements	4030	26	88
PMIDs With 3 Elements	4796	118	1066
PMIDs With > 3 Elements	25,415	39,088	22,159

**Table 3 pharmaceutics-17-00610-t003:** Table summarizing some examples of combinations of ICIs with each category and brief explanation of rationale.

Category	Example of Combination with ICIs	Rationale
Drug Resistance	ATP-binding cassette transporters	Chronic immune evasion can coincide with increased expression of drug-resistance genes. Tumors that acquire resistance to classical agents may still respond to immunotherapies targeting the PD-1/PD-L1 or CTLA-4 pathways
Anti-inflammatory	COX-2 inhibitor	Anti-inflammatory milieu might enhance or occasionally reduce the efficacy of checkpoint inhibitors
Phytogenic agents/ Chinese Herbal	flavonoids, terpenoids	Enhance dendritic cell function or modulate T regulatory cells
Growth Factor Receptors	EGFR, VEGF/VEGFR	Normalize the tumor vasculature and improve immune cell access
Targeted Therapies	BRAF, EGFR, or mTOR inhibitors	Boost tumor antigen release and create a favorable setting for T-cell priming
Chemotherapy Agents	chemotherapy	Chemotherapy-induced cell death can generate antigenic debris, prompting dendritic cell activation, ICIs support T-cell-mediated clearance of residual tumor cells
Tumor Suppressors	BCL-2, BAX, TP53	Combined modulation of cell death pathways and immune checkpoints can lead to robust tumor regression as consequence of Fas ligand and TNF-related apoptosis
Nucleic Acids/ Gene Therapies	siRNA, mRNA vaccines	Enhance T-cell-mediated killing
Drug Delivery/ Advanced Formulations	liposomes, nanoparticles	Improved release of drugs and off-target effects

## Data Availability

The data presented in this study are available in PubMed at https://pubmed.ncbi.nlm.nih.gov/. These data were acquired through PubMed Central (PMC) APIs. The most informative derived datasets obtained through queries of the original downloaded dataset are uploaded in the Appendix A.

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
