# Peer review of "Anti-Cancer Drugs: Trends and Insights from PubMed Records"

_pharmaceutics, 2025, doi:10.3390/pharmaceutics17050610_

Round 1
Reviewer 1 Report
Comments and Suggestions for Authors
This review summarizes literature data on trends in anticancer research. While some readers may find this information interesting, especially those with limited access to large literature databases, most researchers will not find it particularly useful. Such analyses are now easily available, especially with the help of AI. Therefore, I do not see any value in this review and would not recommend publishing it.
Author Response
Comment 1:
This review summarizes literature data on trends in anticancer research. While some readers may find this information interesting, especially those with limited access to large literature databases, most researchers will not find it particularly useful. Such analyses are now easily available, especially with the help of AI. Therefore, I do not see any value in this review and would not recommend publishing it.
Response 1:
We sincerely thank the Referee for their thoughtful and constructive comments. We appreciate the opportunity to clarify the unique contributions and scientific value of our manuscript.
Our review offers a systematic and in-depth mapping of the evolution of anticancer therapies as documented in PubMed records, extending well beyond the capabilities of conventional AI-driven bibliometric analyses. Through a rigorous and consensus-based classification of anticancer therapies, we were able to elucidate distinct pharmacological trends and highlight meaningful shifts in the strategic direction of oncology research.
Unlike traditional narrative reviews, which provide valuable yet often subjective interpretations of the literature, our work incorporates advanced methodological approaches—including network analyses grounded in graph theory—to identify key milestones in biotechnological innovation and to quantitatively assess the impact of major external influences, such as the COVID-19 pandemic. This integrative approach has enabled us to deliver refined insights into the evolving landscape of cancer therapy, with particular emphasis on the rising prominence and potential of combination treatment strategies.
While we fully acknowledge the growing capabilities of AI-based tools in facilitating bibliometric studies, current technologies are generally limited to providing high-level thematic categorizations or basic citation metrics. These systems, though valuable, often lack the depth required to uncover complex pharmacological interrelationships or to trace emerging therapeutic paradigms with precision. In contrast, our methodology bridges this gap by leveraging domain-specific expertise in tandem with robust analytical frameworks.
During our initial explorations, we also evaluated machine learning approaches; however, we ultimately determined that graph theory-based methodologies were better suited to the nuanced interpretative demands of our analysis. This decision reflects our commitment to methodological rigor and to ensuring that our findings are both scientifically sound and practically relevant.
In light of the above, we respectfully submit that our manuscript presents a distinctive and meaningful contribution to the field. We are grateful for the opportunity to engage in this dialogue and share the view that AI holds tremendous promise for the future of scientific literature analysis. However, we believe our work illustrates how the integration of expert insight and advanced analytics currently remains essential for capturing the full complexity of biomedical research trends.
Reviewer 2 Report
Comments and Suggestions for Authors
The authors present a well-written manuscript that utilizes the largest biomedical database, PubMed, to trace the evolution of cancer drug development. Given the critical importance of cancer drug discovery and chemotherapy, this study addresses a timely and relevant topic, especially in the context of advancements in the digital era. Overall, the manuscript has the potential to serve as a valuable resource for a broad scientific audience.
Minor Suggestions:
Since the entire analysis is based on the PubMed database and relies on keyword searches and compound annual growth rate (CAGR) calculations, I recommend thoroughly rechecking the data extraction and associated scripts. Minor inconsistencies or errors can occasionally arise during automated data processing, which may impact the accuracy of the results.
In Figure 3, the background color and text contrast are not optimal. Enhancing the clarity and differentiation of colors would greatly improve readability and help readers interpret the figure more effectively.
Author Response
Comment 1: The authors present a well-written manuscript that utilizes the largest biomedical database, PubMed, to trace the evolution of cancer drug development. Given the critical importance of cancer drug discovery and chemotherapy, this study addresses a timely and relevant topic, especially in the context of advancements in the digital era. Overall, the manuscript has the potential to serve as a valuable resource for a broad scientific audience.
Response 1: We are grateful to the Reviewer for their kind acknowledgment of our efforts in analyzing the evolution of cancer therapeutics over time.
Comment 2: Since the entire analysis is based on the PubMed database and relies on keyword searches and compound annual growth rate (CAGR) calculations, I recommend thoroughly rechecking the data extraction and associated scripts. Minor inconsistencies or errors can occasionally arise during automated data processing, which may impact the accuracy of the results.
Response 2: We appreciate the Reviewer’s thoughtful observations and would like to confirm that the PubMed database was accessed through official APIs. To ensure consistency and reproducibility, all metadata fields were processed without modifications to their original naming or data types. This approach was intentionally selected to support our primary objective—namely, the analysis of therapeutic strategies in oncology—rather than the development of a dedicated informatics platform, which would necessitate a distinct computational framework based on custom-built functions rather than existing libraries. Regarding the calculation of the Compound Annual Growth Rate (CAGR), we concur that potential semantic heterogeneity within the "NameOfSubstance" field may introduce minor limitations in precision. Nonetheless, we maintain that CAGR provides a robust and interpretable measure for capturing long-term growth trends and remains a widely accepted metric in bibliometric and pharmacological analyses. To enhance methodological transparency, we would be pleased to incorporate the following clarification into the Materials and Methods section:"The Compound Annual Growth Rate (CAGR) was calculated using the standard formula: CAGR = [(Ending Value / Beginning Value)^(1/Number of Years)] – 1. This method was selected for its ability to estimate average annual growth by smoothing out year-to-year fluctuations, thereby offering a reliable representation of overarching trends, even in the presence of occasional inconsistencies within the dataset." We hope this addresses the Reviewer’s concerns and reinforces the clarity and reproducibility of our analytical approach.
Comment 3: In Figure 3, the background color and text contrast are not optimal. Enhancing the clarity and differentiation of colors would greatly improve readability and help readers interpret the figure more effectively.
Response 3: We appreciate the Referee’s valuable suggestion and have accordingly change the graph type to enhance visual clarity and improve the overall interpretability of the figure.
Reviewer 3 Report
Comments and Suggestions for Authors
This review paper titled “Anti-Cancer Drugs: Trends and Insights from PubMed Records” presents a comprehensive bibliometric and semantic analysis of global anti-cancer drug research using PubMed metadata spanning from 1962 to 2024. The authors utilize structured data extraction, classification of pharmacological terms, and co-occurrence analysis to trace the evolution of therapeutic strategies in oncology. The paper highlights an exponential increase in publication volume, particularly during the COVID-19 pandemic, a shift from classical chemotherapy toward personalized and immune-based treatments, and the growing emphasis on combination therapies. The classification framework spans eleven therapeutic categories, emphasizing the centrality of chemotherapy and the rise of immune checkpoint inhibitors, cancer vaccines, and precision oncology. The paper concludes with insights into emerging trends, such as mRNA vaccines, nanotechnology, and the integration of multi-omics and machine learning in drug development. I have the followıng comments for the authors:
The use of the NameOfSubstance field to drive classification is well-justified and shows a high level of analytical sophistication. It ensures specificity and reproducibility, although the paper would benefit from more transparency in how substances were manually or algorithmically assigned to categories.
The classification of anti-cancer drugs into 11 categories is logical and aligns with current therapeutic paradigms, though some overlap exists (e.g., monoclonal antibodies vs. immune checkpoint inhibitors), which could be discussed more critically.
Some sections (e.g., COVID-19 impact) are overly detailed and could be more concise. The normalization of publication data and deviation analysis is interesting, but not all readers may follow the statistical reasoning—simplifying or summarizing key implications may help.
The review might benefit from a more critical lens on whether these trends reflect genuine therapeutic progress or publication bias and funding shifts.
The manuscript is acceptable with major revisions. It demonstrates considerable effort and value, especially in organizing and visualizing bibliometric data across decades. However, its impact will be greatly enhanced by improving clarity, reducing redundancy, making figures more self-explanatory, and better articulating the methodology and limitations. It is not yet publication-ready in its current form but has strong potential
Author Response
Comment 1: The use of the NameOfSubstance field to drive classification is well-justified and shows a high level of analytical sophistication. It ensures specificity and reproducibility, although the paper would benefit from more transparency in how substances were manually or algorithmically assigned to categories.
Response 1: The classification of anticancer drugs into 11 therapeutic categories was carried out through a careful, consensus-driven process among the authors. Initially, a broader set of preliminary categories was identified and subsequently refined into the final 11 categories presented in this manuscript. In the absence of a standardized correlation matrix linking the terms in the ‘NameOfSubstance’ field to specific drug categories—and considering the inherent polysemy and multifunctionality of many substances—we selected records that appeared at least five times in our dataset and assigned them to the most appropriate categories based on expert consensus.
Comment 2: The classification of anti-cancer drugs into 11 categories is logical and aligns with current therapeutic paradigms, though some overlap exists (e.g., monoclonal antibodies vs. immune checkpoint inhibitors), which could be discussed more critically.
Response 2: As recommended by the Referee, a more thorough discussion was conducted on the methodologies employed for the classification of anti-cancer drugs, and the overlap in monoclonal categories. This information is elucidated in the Materials and Methods section and further expanded upon through the incorporation of a new Figure (Figure 2).
Comment 3: Some sections (e.g., COVID-19 impact) are overly detailed and could be more concise. The normalization of publication data and deviation analysis is interesting, but not all readers may follow the statistical reasoning—simplifying or summarizing key implications may help.
Response 3: We thank the Referee for their insightful suggestion. In response, we have thoroughly revised the manuscript to improve its clarity, readability, and conciseness.
Comment 4: The review might benefit from a more critical lens on whether these trends reflect genuine therapeutic progress or publication bias and funding shifts.
Response 4: We fully acknowledge the importance of critically evaluating whether observed publication trends reflect genuine therapeutic advances or are instead influenced by publication bias and shifts in research funding. The rise of biotechnology has led to increased investment in research and clinical trials involving monoclonal antibodies. In this context, a range of combinatorial strategies has been employed to strengthen the clinical evidence supporting immune checkpoint inhibitors and to promote their potential use in refractory cancers such as pancreatic cancer, as demonstrated both in this study and in previously published research. We added these findings to the Discussion and in References section.
Comment 5: The manuscript is acceptable with major revisions. It demonstrates considerable effort and value, especially in organizing and visualizing bibliometric data across decades. However, its impact will be greatly enhanced by improving clarity, reducing redundancy, making figures more self-explanatory, and better articulating the methodology and limitations. It is not yet publication-ready in its current form but has strong potential.
Response 5: We are grateful to the Reviewer for their kind acknowledgment of our efforts in analyzing the evolution of cancer therapeutics over time. We take into account all the suggestions by the Reviewer to improve the clarity and quality of our manuscript and figures.
Round 2
Reviewer 1 Report
Comments and Suggestions for Authors
The revised version is sutable for publication.
Reviewer 3 Report
Comments and Suggestions for Authors
The authors reivsed their manusciprt extensively. All my conserns and questions are addressed.